# Physical health of care-experienced young children in high-income countries: a scoping review protocol

Daniel R R Bradford ![ORCID],[1] Mirjam Allik ![ORCID],[1] Alex D McMahon,[2] Denise Brown ![ORCID] [1]

[1]MRC/CSO Social and Public Health Sciences Unit, University of Glasgow, Glasgow, UK
[2]School of Medicine, Dentistry and Nursing, University of Glasgow, Glasgow, UK

**Correspondence to**
Daniel R R Bradford;
d.bradford.1@research.gla.ac.uk

## ABSTRACT

**Introduction** Care-experienced children have poorer health, developmental, and quality of life outcomes across the lifespan compared to children who are not in care. These inequities begin to manifest in the early years. The purpose of the proposed scoping review is to collate and synthesise studies of the physical health of young care-experienced children. The results of the review will help map the distribution of health outcomes, identify potential targets for intervention, and assess gaps in the literature relating to this group.

**Methods and analysis** We will carry out a scoping review of the literature to identify studies of physical health outcomes in care-experienced children. Systematic literature searches will be carried out on the MEDLINE, CINAHL and Web of Science Core Collection databases for items indexed on or before 31 August 2022. Studies will be included where the participants are aged 3 months or greater and less than 6 years. Data elements extracted from included studies will include study objectives, health outcomes, participant demographics, care setting characteristics and bibliographic information. The results of the review will be synthesised and reported using a critical narrative approach. Comparisons between care and non-care populations will be reported if sufficient studies are identified.

**Ethics and dissemination** Data will be extracted from publicly available sources, so no additional ethical approval is required. Results will be published in a peer-reviewed journal article. Furthermore, they will be shared in summary reports and presented to local authorities, care organisations and other relevant stakeholders that can influence healthcare policy and procedure relating to young children in care.

## BACKGROUND

A significant number of infants require support beyond that which their biological parents can provide. To have these additional support needs met, some children come formally under the care of social care services provided by local or national government agencies. This may mean receiving extra input from health and social care workers while remaining in the parental home; living in kinship care; living in foster care; being permanently adopted by non-biological parents or living in residential care homes.

---

**STRENGTHS AND LIMITATIONS OF THIS STUDY**

⇒ The scoping review method facilitates the synthesis of evidence from varied study designs concerning physical health outcomes.
⇒ The prospective review will follow validated methods for literature screening, data extraction and reporting to maximise transparency and replicability.
⇒ Exclusion of grey literature and articles without English-language abstracts are limitations of the prospective review.

---

The proportion of children in care varies internationally due to differing legal and cultural approaches. Examples from high-income countries include 14 per 1000 children in Scotland,[1] 6.7 per 1000 in England[2] and 4.0 per 1000 infants in Sweden.[3] These cross-sectional census figures underestimate the number of children who have ever been in care since there is a constant flux of children into and out of care. Although care-experienced children constitute a small portion of the population of children in high-income countries, these children deserve increased attention due to the additional challenges they face to achieving good health.[4] There is a clear motivation to provide support to improve health outcomes for such children.

Children with care experience have poorer outcomes in many domains of health and well-being compared with their peers in the non-care population.[5–9] Mental ill health is prevalent across the lifespan in this group and has been the subject of many reviews.[10–12] Similarly, developmental delay and disorder in socioemotional, cognitive, and behavioural domains are common in care-experienced children and these topics have been reviewed previously.[12–14] The literature on physical health and physical development problems in care-experienced children is less well established, particularly for those in the early years. Despite physical ill health problems being highly prevalent in this population,[15–18] these

health outcomes have not yet been the subject of a review. Bringing together the results of studies which investigated this topic is therefore a useful exercise and is the purpose of the present work. Our results will help identify common conditions affecting children in care, highlight health conditions and inequalities that may benefit from the development of specific interventions and identify potential gaps in the literature to be addressed.

The results of studies comparing physical health outcomes in care-experienced children and children who have never been in care are less clearly polarised than those for mental health and socioemotional, cognitive, and behavioural development, where the former group are unequivocally worse off.[17 19 20] In the physical domain, there are instances where care-experienced children are observed to have a reduced disease burden, although these are typically for less severe disease. For example, asthma has been found to be less prevalent in samples of children in care than controls from the non-care population.[20 21] By contrast, Turney and Wildeman reported a significantly higher crude proportion of children aged 0–17 years in foster care with asthma.[17] However, after adjusting for birth and socioeconomic factors, the difference in proportion compared with the non-care group became negligible. Fleming *et al* also reported no significant difference in asthma prevalence (after adjustment for birth and socioeconomic factors) in their population-level administrative health data study of school-aged children.[19] Reviewing the available literature can help to clarify such ambiguities. There is, of course, some risk that differences between care-experienced and other children in the prevalence of diagnosed health conditions are due to differences in health-seeking behaviour rather than truly better health. Regardless, improved knowledge about differences in the prevalence of physical ill health between children in care and children in the general population is needed to understand potential associations and/or causes.

Some differences in physical health conditions between children in care and children in the general population are conflicting or ambiguous. However, the subset of conditions caused by, or strongly associated with, abuse and neglect—such as bone fractures and poisonings—are evidently more common in care-experienced populations.[3 22–24] This is unsurprising since abuse and neglect are common reasons for a child becoming formally cared for.[25–27] Although these traumatic experiences have clear and well-documented consequences for socioemotional development,[28] there are also consequences for physical health. Hospitalisations due to both injury and self-harm have been found to be more common in school-aged children in care than in their non-care peers, after adjusting for a range of socioeconomic and maternal factors.[19] Neglectful parenting also has consequences for conditions where proactive health behaviours are vital. For example, dental health problems were significantly more frequently identified in children in care than in the general population during a universal

dental health screening programme in Scotland.[6] This difference remained after adjustment for socioeconomic status and echoes previous studies reporting increased dental health issues in this group.[16 18 21 29] Similarly, care-experienced children are more likely to have incomplete immunisation, or to have been fully immunised at older ages.[16 18] These examples highlight that children in care face unique challenges to their health. These challenges are compounded by factors deleterious to health that are associated with poorer socioeconomic circumstances which care-experienced children more often reside in.[6 30] Identification and mitigation of the consequences of these unique risk factors can help to improve health policy and practices relating to this group.

Infancy and childhood are a period where growth and development are highly sensitive to factors in the individual's environment. Being subject to emotional and physical adversity during this period is associated with long-term negative physical health outcomes.[31 32] If untreated or uncorrected during early childhood, some malleable impairments to health may become permanent. Strabismus (misalignment of the visual axes of the eyes) is an example of such a correctable condition. Intervention to treat this condition in the early years is key to avoiding this becoming a lifelong issue.[33] Additionally, children who enter care early in life also tend to experience different pathways to those becoming cared for during middle childhood or adolescence.[34] Taken together, this suggests children with experience of formal care in the early years may exhibit a distinct cluster of healthcare needs compared to similarly-aged children who are not in care. Furthermore, their healthcare needs likely differ from children who first become cared for later in life. By investigating physical health in the younger care-experienced group in more detail, it is possible that opportunities for early intervention can be better exploited to improve subsequent well-being.

## Study objectives

Previous reviews and meta-analyses have looked at socioemotional and behavioural development,[12 13 28 35] mental health,[10 12 36] effectiveness of interventions and engagement with healthcare providers,[37–39] or have focused on specific subgroups of children with care-experience (e.g., children with prenatal substance exposure,[40] young adults leaving care).[41] Based on preliminary searches of MEDLINE and the Cochrane Database of Systematic Reviews, it appears that there has not been a review of physical health outcomes in young care-experienced children to date. The present review aims to fill that gap. A scoping review is an appropriate methodology for this purpose. This type of review can establish the scope and common themes within existing research into the topic and indicate which physical health conditions are frequently observed in young care-experienced children. This will be of interest to policy-makers and healthcare practitioners.[42–44] Furthermore, under-researched health conditions may be identified by comparing the results of

this review with the more substantial body of literature on the physical health of young children who are not in care.

## Concept definition

This review will address the literature on physical health. We have defined physical health conditions as those primarily rooted or expressed in physiology. While we recognise that psychiatric conditions and socioemotional, cognitive, and behavioural developmental conditions can have organic and/or physiological aspects, these conditions have been previously reviewed (see above). In addition, the proposed review is restricted to children in care in high-income countries. This is because these children face notably different types of physical health conditions to those in low-income and middle-income countries.[45 46]

## METHODS AND ANALYSIS

This review has been designed in line with the Joanna Briggs Institute (JBI) guidelines for carrying out scoping reviews.[47] It is also informed by Arksey and O'Malley's original framework[42] and the additional guidance laid out by Levac, Colquhoun and O'Brien.[43] The five steps of the review procedure of Arksey and O'Malley are detailed below. The JBI framework aligns with the Preferred Reporting Items for Systematic reviews and Meta-Analyses extension (PRISMA-ScR) statement on reporting and conduct of scoping reviews,[48] also used here.

### Stage 1: identifying the research question

The Population, Concept, and Context (PCC) framework was used to define the review question. Elements of the PCC are described in detail in table 1. This review intends to primarily answer the question: 'Which physical health conditions have been studied or observed to affect young care-experienced children in high-income countries?' If there are sufficient relevant primary studies which compare care-experienced children to control groups in the general, non-care population, then the review will also address the question: 'How do these health conditions affect children in care compared to their peers in the general population?'

### Stage 2: identifying relevant studies

Keywords and synonyms to identify (1) care-experienced children; (2) health and wellbeing and (3) children aged greater than or equal to 3 months and less than 6 years will be combined to search the bibliographic databases. Studies of the health of care-experienced children previously known to the authors were used to identify keywords to form the basis of the search strategy for this review. Iterative pilot searches of MEDLINE were carried out to identify additional relevant keywords. This process ended when saturation was reached, and additional keywords/variants did not bring forth additional search results. Advice on keywords and search strategy was also taken from an Information Scientist within the University of Glasgow MRC/CSO Social and Public Health Sciences Unit. These keywords are detailed in an online supplemental file. The review will search for relevant studies indexed by the MEDLINE, CINAHL and Web of Science Core Collection (Science Citation Index (SCI-EXPANDED) and Social Sciences Citation Index (SSCI)). Search strategies are provided in the an online supplemental file 1. The search strategy will be adapted appropriately for the remaining databases. Following searches, all results will be imported to Rayyan and duplicates removed.[49]

### Stage 3: study selection

Search results will be screened for relevance to the Population, Concept, and Context described above. More specific inclusion and exclusion criteria are described below. Screening will be carried out using Rayyan.[49] Screening will take place in two stages: title and abstract screening then full texts. All titles and abstracts will be screened by DRRB and a minimum of 20% will be screened independently by a second reviewer. If it is unclear whether an article should be included based on the details in the title/abstract and there is potential for it to be relevant, then it will be progressed for full-text screening (e.g., if the study clearly includes care-experienced children, but the age of participants is not immediately apparent, it will be progressed to the full-text review stage). Non-English-language articles with

**Table 1** Detailed description of the Population, Concept and Context used a foundation for the review research questions

| | |
|---|---|
| Population | Care-experienced children who meet all following criteria:<br>► Aged greater than or equal to 3 months and less than 6 years<br>► Have experience of formal care settings listed in the introduction of this protocol, or similar settings<br>► Not adopted during the first 3 months of their life<br>► Not cared for in healthcare facilities, to avoid confounding |
| Concept | Physical health outcomes will be considered as those health conditions which are primarily rooted in or expressed through physiology. As a guide, these conditions will be those that are not listed in the International Classification of Diseases (11th Revision)[53] diagnostic manual top-level entity on *Mental, behavioural or neurodevelopmental disorders*. |
| Context | Included studies will be restricted to those carried out in high-income countries. This is because children in care in low-income and middle-income countries face significantly different physical health issues.[45 46] |

English-language abstracts that indicate relevance to this review will be translated and included. Crude inter-rater agreement will be reported along with Cohen's Kappa[50] and Scott's Pi.[51] If crude agreement falls below 95% in this sample of 20% of titles/abstracts, then all items will be screened by at least two reviewers. During the next stage of screening, all full texts will be screened by DRRB and at least 20% will be screened by a second reviewer. Crude inter-rater agreement will be reported along with Cohen's Kappa and Scott's Pi. If crude agreement falls below 95%, then all full texts will be screened by at least two reviewers. Disagreements at any stage of screening will be resolved by discussion between the two reviewers. If consensus is not achieved, a third reviewer will make the final decision. Reasons for exclusion will be recorded and documented at each stage and reported in line as per the PRISMA-ScR guidelines.[48]

### Inclusion criteria

► Included studies will consider children currently living in family care with additional formal support from social or child protection services, kinship care, foster care, residential care, as well as children living with adopted families. Studies of children with previous experience of these settings will also be included, as will studies which aggregate groups of children either currently in or having previously been in care.

► Studies will be included if all children are aged greater than or equal to 3 months and less than 6 years, or if subgroup data for this group are reported.

► Study outcomes must include physical health conditions or diagnoses by a health professional, diagnostic manual, or validated measure(s). Studies which have made minor modifications to validated instruments may be included at reviewers' discretion. Studies which use parental report of a current or previous diagnosis by a clinician will be included.

► Studies will be included if they were carried out in high-income countries as defined by the World Bank.[52]

► Only articles and reports published in peer-reviewed journals will be included.

### Exclusion criteria

► Studies of children living in healthcare settings will be excluded to avoid confounding.

► Papers that look at only physiological markers will be excluded. For example, studies of expression of hormones, such as salivary cortisol, will be excluded unless explicitly linked to a health condition.

► Articles without an English language title and abstract.

### Stage 4: charting the data

A bespoke data charting tool has been developed following identification of key variables from relevant studies identified during pilot searches (see the online supplemental file for details). The data to be extracted include information about the sample demographics (e.g., age and gender balance), type of care placement or setting, health and developmental outcomes considered, study design, and country in which the study was carried out. Bibliographic information will also be included. Data extraction from included full-text articles will be carried out by one reviewer (DRRB) and at least 20% of these extractions will be verified independently by a second reviewer.

### Stage 5: collating, summarising and reporting the results

The results of the search and screening process will be reported using a PRISMA-ScR flow diagram.[48] The details of included studies will be presented as tables. Aggregated descriptive statistics about studies will be presented. A thematic analysis will be carried out to identify emergent clusters of health conditions (if sufficient studies are identified). Differences in outcomes between children with and without care experience will be summarised if sufficient comparative studies are identified. Bibliographic information will be collated and presented including title, author(s), year of publication, and publishing journal.

## ETHICS AND DISSEMINATION

This study will review publicly available knowledge, so requires no additional ethical approval. It is the intent of the authors to present the findings of this review as an article in a peer-reviewed journal and at relevant conferences. The results of this review will also inform future research into health conditions that may disproportionately affect care-experienced children, as well as potentially stimulating research into health conditions which have been under-researched in this group. This review also serves as a backdrop to a planned research project by the authors into the health and development of preschool children in care in Scotland. Additionally, the findings of this review will be of interest to practitioners and policymakers working to understand and improve the health outcomes of care-experienced children. Our results will therefore also be circulated to interested stakeholders to make a direct and immediate contribution to the field.

**Acknowledgements** DRRB would like to thank Valerie Wells (Information Scientist, MRC/CSO Social & Public Health Sciences Unit) for advice about search strategies, and Avril Mason (Consultant Paediatric Endocrinologist, NHS Greater Glasgow and Clyde) for reviewing an initial draft of the manuscript.

**Contributors** DRRB conceptualised the scoping review and research questions, developed the search strategy and drafted the manuscript. DB, MA and ADMcM critically reviewed the review methods and provided comments on all aspects of initial draft of the manuscript. All authors have made substantial intellectual contributions to the work.

**Funding** This work was funded as by the Medical Research Council and the Scottish Government Chief Scientist Office as part of an MRC PhD studentship awarded to DRRB (MC_ST_00022). MA/DB were supported by the Economic and Social Research Council (grant number ES/T000120/1), the Medical Research Council (MC_UU_00022/2) and the Scottish Government Chief Scientist Office (SPHSU17).

**Competing interests** None declared.

**Patient and public involvement** Patients and/or the public were not involved in the design, or conduct, or reporting or dissemination plans of this research.

**Patient consent for publication** Not required.

**Ethics approval** Not applicable.

**Provenance and peer review** Not commissioned; externally peer reviewed.

**Data availability statement** Data sharing not applicable as no data sets generated and/or analysed for this study.

**ORCID iDs**
Daniel R R Bradford http://orcid.org/0000-0002-7523-8764
Mirjam Allik http://orcid.org/0000-0003-1674-3469
Denise Brown http://orcid.org/0000-0002-5195-5312

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
