## [Reviewer comments · BMJ Open]

ARTICLE DETAILS

TITLE (PROVISIONAL)	Physical health of care-experienced young children in high-income countries: a scoping review protocol
AUTHORS	Bradford, Daniel; Alik, Mirjam; McMahon, Alex; Brown, Denise

VERSION 1 – REVIEW

REVIEWER	Gardiner, Julian University of Oxford, Education
REVIEW RETURNED	28-Jun-2022

GENERAL COMMENTS	This is a well written protocol addressing an important area of research. Minor corrections P 7, line 42 “...and children the general population...” Change to “...and children in the general population...” P 9, lines 52-55 “The proposed review is restricted to children in care in low- and middle-income countries.” As it stands, this contradicts what is stated in the rest of the protocol. This should be corrected, e.g. to: “The proposed review excludes children in care in low- and middle-income countries.” P 14, lines 48-51 “It is the intent of the authors to present the findings of this review as an article in peer-reviewed journal and at relevant conferences.” Change to: “It is the intent of the authors to present the findings of this review as an article in a peer-reviewed journal and at relevant conferences.”
---

REVIEWER	Alderson, Hayley
-----------------	------------------

	Newcastle University
REVIEW RETURNED	12-Jul-2022

GENERAL COMMENTS	Thank you for this interesting protocol considering an important subject area. A few minor comments from me that need addressing as follows: Page 6 line 12- please can you explain what a local authority is- this may not be clear to audiences outside of the UK context. Page 9 line 12- you state developmental- please elaborate, developmental delay/developmental challenges? Page 9 line 52- You state "the proposed review is restricted to children in low- and- middle- income countries. However in table 1, you state "included studies will be restricted to those carried out in high- income countries". Please could you rectify this and make it clear and consistent which population you are focusing on. Page 13 line 27- you state "only peer reviewed reports" will be included- Please could you elaborate on how you will identify if a report has been peer reviewed? or whether you are only including peer reviewed academic articles.
--

REVIEWER	Mason, Kate The University of Melbourne
REVIEW RETURNED	16-Jul-2022

GENERAL COMMENTS	Strengths and Limitations:  - Typo second last bullet point – ‘out’ should be ‘our’ - If restricting to English, please explain the implications. But please also consider expanding to include other languages. If not many, translation may be feasible. The protocol later states that “Articles without an English language title and abstract” will be excluded. Does this mean if the full article is not in English but the title and abstract have been translated, then translation of the full article would be sought? Concept definitions:  - Page 9, line 52 states “the proposed review is restricted to children in care in low- and middle income countries.”. This seems to be an error – the protocol describes the review limited to high-income settings. Study selection:  - It’s unclear how studies with a focus on developmental outcomes will be treated. Early on the authors note that these have already been the subject of reviews, yet the term developmental appears a couple of times alongside physical health outcomes (e.g. in search strategy). - If I understand correctly, I think the review will be limited to studies of health in children under 6 (as well as the period of care have to occur in that age range) but please clarify that studies of health outcomes measured later in childhood or beyond are out of
---

	scope, if this is that case. Related to this, there are implications for understanding causal pathways if care experience and health outcomes are defined/measured contemporaneously – poor health outcomes could be due to pre-care or in-care exposures. Does this matter for the purposes of this review? Stage 5: Collating, summarising, and reporting the results:  - On Page 14 the authors state that “A thematic analysis taking a descriptive-analytical stance will be carried out”. In the Abstract the approach to be taken is described as a “critical narrative approach”. I don’t have the expertise to be sure, so please just confirm that these descriptions are compatible, and perhaps expand briefly for clarity. Search strategy:  - Are the authors confident that the age-related search terms won’t exclude studies of children aged 5? The search terms mostly focus on under 5 and preschool children, yet the study population of interest is defined as children under 6.
--	--

VERSION 1 – AUTHOR RESPONSE

Reviewer comments

Reviewer 1

#	Comment	Response
4	“...and children the general population...” change to “...and children in the general population...”	Missing word inserted.
5	“The proposed review is restricted to children in care in low- and middle-income countries.” As it stands, this contradicts what is stated in the rest of the protocol. This should be corrected, e.g. to: “The proposed review excludes children in care in low- and middle-income countries.”	Wording corrected to clarify that the review will be restricted to high-income countries only.
6	“It is the intent of the authors to present the findings of this review as an article in peer-reviewed journal and at relevant conferences.” Change to: “It is the intent of the authors to present the findings of this review as an article in a peer-reviewed journal and at relevant conferences.”	Missing word inserted.

Reviewer 2

#	Comment	Response
7	Please can you explain what a local authority is- this may not be clear to audiences outside of the UK context.	Relevant sentence edited to “To have these additional support needs met, some children come formally under the care of local authorities or social care services provided by local or national government agencies.”
8	You state developmental- please elaborate, developmental delay/developmental challenges?	We have added additional details throughout the manuscript where “development” is used. These details help to highlight the types of developmental outcomes or challenges being referred to (e.g., by adding explicit reference to socioemotional, behavioural domains etc.). Hopefully, these additions make clear the intent to exclude research which focuses on socioemotional, cognitive, and behavioural development (both delayed and typical) while including developmental outcomes that relate to physical health and development (e.g., stature, motor control). We welcome further comments from reviewers if this remains ambiguous.
9	You state "the proposed review is restricted to children in low- and- middle- income countries. However in table 1, you state "included studies will be restricted to those carried out in high- income countries". Please could you rectify this and make it clear and consistent which population you are focusing on.	As per our response to Comment #5; the wording corrected to clarify that the review will be restricted to high-income countries only.

10	You state "only peer reviewed reports" will be included- Please could you elaborate on how you will identify if a report has been peer reviewed? or whether you are only including peer reviewed academic articles.	The inclusion criterion has been rephrased for clarification that only articles and reports published in peer-reviewed journals will be included. Our use of the word “reports” was to represent technical notes, short reports etc., as opposed to reports published by organisations without external peer review. New phrasing of inclusion criterion: “Only articles and reports published in peer-reviewed journals will be included.”
----	---	---

Reviewer 3

#	Comment	Response
11	Strengths and Limitations - Typo second last bullet point – ‘out’ should be ‘our’	Typo corrected.
12	If restricting to English, please explain the implications. But please also consider expanding to include other languages. If not many, translation may be feasible. The protocol later states that “Articles without an English language title and abstract” will be excluded. Does this mean if the full article is not in English but the title and abstract have been translated, then translation of the full article would be sought?	The inclusion criterion has been adjusted slightly and clarification added. As far as is possible, we will now include and translate non-English-language papers that have an English-language abstract which suggests the paper meets the inclusion criteria. Any deviations from this will be mentioned in the resulting review article, i.e., in the case of being unable to translate an article deemed to meet the inclusion criteria. Sentence added to method section to reflect this: “ Non-English-language articles with English-language abstracts that indicate relevance to this review will be translated and included. ”
13	[The manuscript] states “the proposed review is restricted to children in care in low- and middle income countries.”. This seems to be an error – the protocol describes the review limited to high-income settings.	Wording corrected to clarify that the review will be restricted to high-income countries only.

14	It's unclear how studies with a focus on developmental outcomes will be treated. Early on the authors note that these have already been the subject of reviews, yet the term developmental appears a couple of times alongside physical health outcomes (e.g. in search strategy).	Please see the response to comment #8. Re. search strategy: Some studies that look at childhood development will include developmental outcomes that are physical in nature (e.g. stature, motor control). As such, "Child Development" was included as a keyword/MeSH in the search strategies to maximise the pool of potential search results.
15	On Page 14 the authors state that "A thematic analysis taking a descriptive-analytical stance will be carried out". In the Abstract the approach to be taken is described as a "critical narrative approach". I don't have the expertise to be sure, so please just confirm that these descriptions are compatible, and perhaps expand briefly for clarity.	The mention of "descriptive-analytical stance" has been removed for brevity and clarity.
16	Are the authors confident that the age-related search terms won't exclude studies of children aged 5? The search terms mostly focus on under 5 and preschool children, yet the study population of interest is defined as children under 6.	Search terms updated. The reference to under-fives is a remnant of a previous iteration of the age-related inclusion criterion.

VERSION 2 – REVIEW

REVIEWER	Alderson, Hayley Newcastle University
REVIEW RETURNED	11-Aug-2022

GENERAL COMMENTS	Thank you for addressing the reviewers comments, the paper has been strengthened because of this. A very important area of research.
--

REVIEWER	Mason, Kate The University of Melbourne
REVIEW RETURNED	19-Aug-2022

GENERAL COMMENTS	All reviewer comments have been addressed and I am happy to recommend this be accepted for publication.
---